# Rational Design of Self-Emulsifying Pellet Formulation of Thymol: Technology Development Guided by Molecular-Level Structure Characterization and Ex Vivo Testing

**DOI:** 10.3390/pharmaceutics14081545

**Published:** 2022-07-25

**Authors:** Jan Macku, Katerina Kubova, Martina Urbanova, Jan Muselik, Ales Franc, Gabriela Koutna, Miroslava Pavelkova, David Vetchy, Josef Masek, Eliska Maskova, Jiri Brus

**Affiliations:** 1Department of Pharmaceutical Technology, Faculty of Pharmacy, Masaryk University Brno, 612 00 Brno, Czech Republic; mackuj@mail.muni.cz (J.M.); muselikj@pharm.muni.cz (J.M.); franca@pharm.muni.cz (A.F.); 507108@mail.muni.cz (G.K.); pavelkovam@pharm.muni.cz (M.P.); vetchyd@pharm.muni.cz (D.V.); 2Department of NMR Spectroscopy, Institute of Macromolecular Chemistry, Czech Academy of Sciences, 162 06 Prague, Czech Republic; urbanova@imc.cas.cz; 3Department of Pharmacology and Toxicology, Veterinary Research Institute, 621 00 Brno, Czech Republic; masek@vri.cz (J.M.); maskova.e@vri.cz (E.M.)

**Keywords:** thymol, solid-state NMR, self-emulsifying pellet, rational design, structure, ex vivo testing

## Abstract

The growing need for processing natural lipophilic and often volatile substances such as thymol, a promising candidate for topical treatment of intestinal mucosa, led us to the utilization of solid-state nuclear magnetic resonance (ss-NMR) spectroscopy for the rational design of enteric pellets with a thymol self-emulsifying system (SES). The SES (triacylglycerol, Labrasol^®^, and propylene glycol) provided a stable o/w emulsion with particle size between 1 and 7 µm. The ex vivo experiment confirmed the SES mucosal permeation and thymol delivery to enterocytes. Pellets W90 (MCC, Neusilin^®^US2, chitosan) were prepared using distilled water (90 g) by the M1–M3 extrusion/spheronisation methods varying in steps number and/or cumulative time. The pellets (705–740 µm) showed mostly comparable properties—zero friability, low intraparticular porosity (0–0.71%), and relatively high density (1.43–1.45%). They exhibited similar thymol release for 6 h (*burst effect* in 15th min ca. 60%), but its content increased (30–39.6 mg/g) with a shorter process time. The M3-W90 fluid-bed coated pellets (Eudragit^®^L) prevented undesirable thymol release in stomach conditions (<10% for 3 h). A detailed, ss-NMR investigation revealed structural differences across samples prepared by M1–M3 methods concerning system stability and internal interactions. The suggested formulation and methodology are promising for other lipophilic volatiles in treating intestinal diseases.

## 1. Introduction

Phytochemicals are used in current pharmacotherapy only to a limited extent, although many significant benefits, such as fewer adverse effects, simplifying clinical trials, reducing antibiotic resistance, and environmental contamination by chemical compounds could be expected. In the COVID-19 pandemic context, much attention is being paid to a rigorous mapping of natural medicines’ biological actions, including terpenic substances expressing a broad spectrum of biological activities [1]. Recent comprehensive data [1,2] reveal their potential and new possibilities for their application in pharmacotherapy; for instance, terpenic substances abound with great potential for the treatment of gastrointestinal diseases and digestive disorders, which are widespread and could affect the whole gastrointestinal tract (GIT) [3]. Current treatment consists mainly of “gold standard” chemical drugs, such as antibiotics, chemotherapeutics, corticosteroids, antiseptics or disinfectants, and newly by biologicals in treating inflammatory bowel diseases (IBDs). The main limitations are significant local and systemic adverse effects and the lifelong treatment, including a remission period of inflammatory bowel diseases [4]. Together, these aspects significantly decrease patient adherence to the treatment [5].

The well-known plant compound, thymol, has been extensively investigated. It is a monoterpene phenol, chemically known as 2-isopropyl-5-methyl phenol. Thymol is naturally occurring in many plants, mainly of the family *Lamiaceae*. It exhibits several remarkable biological activities, such as antimicrobial, anti-inflammatory, antioxidant, and many others [6,7]. Due to the unique complex effect, thymol is a potential candidate for treating inflammatory or infectious diseases of the small intestine and colon mucosa [8,9,10,11]. The fundamental problem is that monoterpene molecules are predominantly hydrophobic, with poor or unknown oral bioavailability [12,13]. Pharmacokinetic data for thymol are very limited. According to available literature, thymol is only partially absorbed mainly from the stomach, resulting in a 16% bioavailability. Since free thymol was not detected in plasma, so the bioavailability was set by quantifying metabolized thymol conjugates [14]. Thymol is cumulated in a higher amount in the intestinal mucosa and inner part of GIT, as shown in healthy volunteers [15]. These thymol’s pharmacokinetic characteristics and the proven in vivo efficacy against IBDs predetermine its use in treating intestinal diseases [8,9,11].

Developing a gastro-resistant solid self-emulsifying system is a valuable pharmaceutical technology option to promote thymol penetration into the intestinal mucosa, reducing its undesirable absorption in the stomach and increasing its potential use in pharmacotherapy [16,17]. The coated self-emulsifying pellets thus make it possible to control the release of the active substance and overcome the disadvantages of liquid medicines. Higher stability, taste masking, easier handling, simpler dosing, and increased patient compliance are the main expected benefits that make them great interest to the pharmaceutical industry [18,19]. According to the scientific literature, the coated self-emulsifying pellets represent an alternative to cellulose derivative-based (ethyl cellulose, hypromellose, or hypromellose phthalate) microparticles containing thymol prepared by solvent evaporation or spray-drying methods. Compared to microparticles, they provide many advantages, such as the drug in dissolved form, polar solvents absence, high yield, industrial applicability, low equipment requirements, and in some cases, regular shape. Disadvantages can be seen mainly in the multi-step preparation method [15,20].

The extrusion/spheronisation process involves the incorporation of liquid components (a drug dissolved in the mixture of oil, surfactant, and co-surfactant/cosolvent) into a solid adsorbent (e. g., mesoporous silica (nano)particles), the addition of other solid excipients (with regard to the production or characteristics of pellets), the application of a liquid wetting agent, and finally the coating process [21,22]. During these stages, liquid and solid phases are in temporary or permanent contact. It opens up the scope for interactions, transformations, and transfer of individual components within the dosage form. Consequently, the sophisticated and tightly controlled development and the full exploitation of these systems thus require their precise characterization. It is, however, a stringent requirement, as these systems naturally exist at the borderline between solids and liquids for which the high-quality diffraction data are inherently unavailable. In this respect, ss-NMR has evolved into a potent tool providing precise atomic-resolution structural descriptions of a wide range of complex multi-component pharmaceutical solids and particulate systems [23,24,25,26].

In this context, we have developed an easy-to-implement procedure to explore the molecular-level architecture of the active compounds in *liquisolid* drug delivery systems based on mesoporous silica particles. This way, a unique, previously unknown organogel phase was identified [23]. Overall, we believe that developing new experimental strategies for describing structures of complex drug-delivery systems is vital for the full exploitation and advancement of a novel formulation of active pharmaceutical ingredients.

The presented study thus demonstrates a new approach to the rational design of gastro-resistant pellets loaded with the self-emulsifying system containing thymol. Specifically, the aim of our research was to develop a self-emulsifying pellet formulation of thymol for the therapy of intestinal mucosa inflammation, where the design and optimization of the technology were guided by structural characterization at the molecular level and ex vivo testing. The detailed structural analysis helped us to understand the relationships and extent of interactions in these dosage forms based on the acquired knowledge. In this way, we were able to correlate their targeted formulation with optimized properties. The proposed research and obtained findings thus open new gates for rational design and reproducible preparation of novel mucosal drug delivery systems.

## 2. Materials and Methods

The development of the self-emulsifying pellet formulation of thymol presented in this contribution followed an approach known as *rational design*, where optimization of individual technological parameters was guided by ss-NMR spectroscopy structural characterization and ex vivo testing (see Figure 1). Description of all materials and chemicals used to prepare the self-emulsifying pellet systems and a detailed description of all the applied characterization methods and tests can be found in the sections below.

### 2.1. Materials

Active substance thymol was purchased from Sigma Aldrich (Taufkirchen, Germany). Medium-chain triglycerides (TAG; Fagron, Olomouc, Czech Republic), caprylocaproyl polyoxyl-8 glycerides—Labrasol^®^ (Gattefosse SAS, Saint-Priest, France), and propylene glycol (Dr. Kulich Pharma, Hradec Králové, Czech Republic) were used as starting materials for a liquid self-emulsifying system (SES). In pellet formulations, Neusilin^®^ US2 (NEU; Fuji Chemicals, Tokyo, Japan), chitosan (JBICHEM, Shanghai, China), and microcrystalline cellulose (MCC) (Avicel^®^ PH 101, FMC, Cork, Ireland) served as SES adsorbent, mucoadhesive biopolymer, and a spheronisation enhancer, respectively. A water dispersion of gastro-resistant polymer Eudragit^®^ L30 D-55 (Evonik, Essen, Germany) in combination with a plasticizer triethyl citrate (Sigma-Aldrich, St. Louis, MO, USA) and distilled water (Ph. Eur. 10) was used for pellet coating. All used ingredients were of pharmaceutical grade.

### 2.2. Preparation and Evaluation of Liquid SES

Thymol in 6 g was dissolved in 11.7 g TAG. Labrosol^®^ (19.5 g) was separately mixed with propylene glycol (7.8 g) using a magnetic stirrer (Fisherbrand, Pittsburgh, PA, USA) for 30 s. at 50 rpm. Thymol oil solution was continuously added during stirring, and the mixture was homogenized for another 3 min. The final SES in a total amount of 45 g contained 13.266% of thymol. A fluorescent dye coumarin-6 in 0.066% concentration was added to the SES. After dissolving, the SES with coumarin-6 was mixed with 99 g of distilled water to create spontaneously an emulsion which was observed using a confocal microscope (Leica SP8, Leica Microsystems’, Wetzlar, Germany).

### 2.3. Evaluation of the Emulsion Particle Size and Short-Term Particle Stability

Confocal microscopy was used to evaluate the size of emulsion particles. About 0.5% coumarin-6 was used as a fluorescent marker to visualize the particles. The solution of emulsion particles was prepared as described above. About 200 uL of the final solution was put into µ-Slide 8 Well Chamber (IBIDI) and observed with a 63× objective using 470 excitation laser line (white-light laser) and emission wavelengths of 480–530 nm. The size of particles was observed immediately after mixing SES with distilled water and after 1 h.

### 2.4. Preparation of Uncoated SES Pellets

The wet mass for extrusion (Table 1) was prepared using the universal electronic machine Stephan UMC 5 (A. Stephan u. Söhne GmbH & Co., Hameln, Germany), operating at 1500 rpm.

The method 1 (M1) steps (see Figure 2) were as follow: SES was added to NEU in thirds evenly for 1 min, and each SES addition was followed by a minute of mixing (final time—6 min). After SES adsorption into the NEU structure, MCC and chitosan were added, and the powder mixture was homogenized for the next 1 min. The wetting agent differing in type (water, 0.25% acetic acid, or 0.5% acetic acid) and amount (90, 100, or 110 g) was added in the same manner as the drug-loaded SES. The process was finished with one extra minute of mixing. The M1 wetted mass was extruded in one axial screw extruder (Pharmex 35T, Wyss & Probst, Ettlingen, Germany) with a die plate (perforation diameter 0.78 mm) at a constant speed of 110 rpm. The obtained extrudate was manually transferred on a radial plate (23 cm in diameter) of the spheronizer (Pharmex 35T, Wyss & Probst, Ettlingen, Germany). The spheronizer operated at a rotating speed of 1000 rpm for 2 min. Pellets were placed on a tray and dried in a hot air dryer (Horo, Type 38A, Ostfildern, Germany) at 40 °C for 12 h.

The selected sample was also prepared by M2 and M3 to simplify the preparation process (see Figure 2). The M2 steps were as follows: the powder excipients NEU, chitosan, and MCC were mixed for 3 min. An emulsion formulated by mixing the SES and water was added in quarters evenly for 1 min, and each emulsion addition was followed by a minute of mixing (final time—8 min). The last 1 min mixing finished the process. The M3 has been further simplified: the powder excipients (NEU, chitosan, and MCC) were mixed for 1 min. The whole amount of the emulsion was added evenly for 2 min, followed by 2 min of homogenization (final time—4 min). The last 30 s of mixing finished the process.

### 2.5. Pellet Coating

The dried pellet cores of the selected sample underwent sieve analysis (AS200 Basic, Retsch GmbH & Co., Haan, Germany), and a size fraction of 0.5–0.8 mm was used for the coating. A batch of pellet cores (140 g) was coated in a Wurster-type fluid bed coating unit (Medipo, M-100, Havlickuv Brod Czech Republic). The coating dispersion was prepared by mixing Eudragit^®^ L30 D-55, triethyl citrate, and water (25.78, 2.32, 71.9%). Afterwards, the dispersion was stirred for five hours and filtered through a 0.125 mm sieve. The following conditions for coating were applied: spraying nozzle diameter 1.2 mm, rate of spraying 2.5 mL/min, inlet air temperature 50 °C, atomising air pressure 1.0 bar, drying air volume 25–40 m³/h. The final coating represented 50% of the core pellet mass.

### 2.6. Evaluation of Uncoated SES Pellets

The obtained pellet samples were evaluated for their size and size distribution by a sieve analysis (Retsch, AS 200, Haan, Germany) using a sieve shaker and stainless-steel mesh sieves (125–2000 mm). The selected 0.5–0.8 mm fractions were characterized for their technological properties (fraction yield, sphericity, average diameter, flow properties, pycnometric density, intraparticular porosity, friability, and thymol content, and in-vitro dissolution behavior) to find out the optimal formulation set. Pellet sphericity and average diameter (in 200 randomly selected pellets scanned at magnification ×0.75) were determined by microscope Nikon SMZ 1500 (Nikon, Tokyo, Japan), mounted with digital camera 70AUC02 USB (The Imaging Source, Bremen, Germany), and connected to PC, assessed by software NIS-Elements AR 4.0 software (Nikon, Tokyo, Japan). Hausner ratio (HR), indicating pellet flow properties, was evaluated in triplicate (Ph. Eur. 10). Pycnometric density of final pellet samples and SES powder blends (*n* = 3; Helium pycnometer, Pycnomatic-ATC, Porotec GmbH, Haan, Germany) served as initial data for the calculation of pellet intraparticular porosity. Mechanical properties were evaluated via friability of the 10 g pellet sample (*n* = 1; Erweka TAR 10, Ensenstam, Germany) after 10 min using a stainless-steel drum with 200 glass balls. The HPLC (*n* = 3; Agilent 1260, Agilent Technologies, Santa Clara, CA, USA) method assessed thymol content in the pellets with conditions as follows. A 50 mL of distilled water and 100 mg of the crushed sample were filled into a 100 mL volumetric flask. Flask was ultra-sonicated (Bandelin Sonorex, Berlin, Germany) for 10 min. The contents of the flask were topped up to the mark with methanol. The volumetric flask was shaken, allowed to stand for 24 h, and filtered through a 0.45 µm pore-sized membrane filter into a vial. LiChrospher^®^ 100 RP-18 column (particle size 5 μm) was used for separation. The mobile phase consisted of 50% of acetonitrile and 50% of phosphoric acid (0.02 M), the flow rate of 1.0 mL/min, and the column temperature was 30 °C. Spectra were recorded at a wavelength of 274 nm. The amount of 10 μL of each sample was analyzed. Thymol quantification was based on the linearized calibration curve (R^2^ ≥ 0.99).

Moreover, four-month stability studies were carried out in the selected pellet sample (M1-W90). After qualitative evaluation, plastic cups with pellets in a plastic resealable were placed in stability boxes (Binder, Tuttlingen, Germany) under 25 °C/relative humidity (RH) 60% and 40 °C/RH 75 %, respectively. The crucial points of stability testing were thymol content (*n* = 3), friability (*n* = 3), and dissolution behavior (*n* = 3).

### 2.7. In Vitro Dissolution Study of Uncoated Pellets

Thymol release from uncoated pellets was determined using the dissolution apparatus (Sotax AT-7 Smart, Sotax, Allschwil, Switzerland). The dissolution testing of the 0.5 g pellet sample (*n* = 3) was performed in 500 mL of 6.8 phosphate buffer, enriched with 1% *w*/*w* polysorbate 80, representing the small intestine environment using the paddle method apparatus (100 rpm, 37.0 ± 0.5 °C). The samples were withdrawn after 30, 60, 120, 180, 240, and 360 min and subsequently analyzed for thymol content by HPLC using the same analytical approach mentioned above. Results (in %) were expressed as mean values of released thymol and standard deviation (SD). An influence of type and amount of wetting agent and used method (M1–M3) was assessed by well-known similarity factor (*f*_2_) analysis.

### 2.8. In Vitro Dissolution Study of Coated Pellets

The dissolution study of thymol-coated pellets (1 g, *n* = 3) was performed under the same condition as mentioned. Still, a dissolution medium with varying pH values was used to mimic an intended animal model’s (rabbit) GIT condition. The artificial gastric juice without pepsin of pH 1.2 (1 g of NaCl and 40 mL of HCl in 500 mL of aqueous solution) was used for the initial part (3-h interval) to simulate stomach conditions. It was followed by pH switching to 6.8 (small intestine) by adding Na_3_HPO_4_.12H_2_O (6.25 g/500 mL) for the remaining testing time (6 h). The samples were withdrawn after 60, 120, and 180 min (pH = 1.2), 195, 210, 240, 300, 360, 420, and 540 min (pH = 6.8) and analyzed for thymol content. Results (in %) were expressed as mean values of released thymol and SD.

### 2.9. Scanning Electron Microscopy

The morphology of selected samples was examined by scanning electron microscope (SEM, MIRA3, Tescan Orsay Holding, Brno, Czech Republic) equipped with a secondary electron detector (SED) operating at an accelerating voltage of 3 kV. Cross-sections of the pellets were obtained by cutting the pellet with a razor blade. The sample was fixed to the SEM stage using conductive carbon double-faced tape (Agar Scientific, Essex, UK) and subsequently coated with a 20 nm layer of gold by the argon atmosphere metal sputtering method (Q150R ES Rotary-Pumped Sputter Coater/Carbon Coater, Quorum Technologies, Laughton, UK).

### 2.10. Solid-State NMR Spectroscopy

Solid-state NMR spectra were measured at 11.7 T using a Bruker Avance III HD 500 WB NMR spectrometer (Karlsruhe, Germany, 2013) with a 3.2 and 4 mm probe head in ZrO_2_ rotors. The ^13^C MAS and CP/MAS NMR spectra employing cross-polarization (CP) were acquired using the standard pulse scheme at a spinning frequency of 5–10 kHz. The optimized recycling delay was 2 s, and the cross-polarization contact time was 2 ms. The strength of the spin-locking fields had a B1(^13^C) field nutation frequency of 62.5 kHz. The number of scans was 6–15 k. ^1^H MAS NMR spectra were acquired using the standard pulse scheme with the repletion delay of 4 s and the number of scans of 4. The 2D ^1^H-^1^H NOESY MAS NMR correlation spectra were measured using the standard three-pulse (NOESY-type) pulse sequence at a spinning frequency of *ω_r_*/2π = 10 kHz and recycle delay of 2 s. The *t*_1_ evolution period consisted of 256 increments, each of 64 scans. Frictional heating of the spinning samples was compensated by active cooling, and the temperature calibration was performed with Pb(NO_3_)_2_.

### 2.11. Ex Vivo Passive Permeation of Coumarin-6 into the Intestinal Epithelium

The passive permeation of coumarin-6 formulated into thymol-containing SES as a model hydrophobic active ingredient was observed using freshly excised rabbit small intestine. The rabbits were purchased from AnLab, s.r.o., Prague, Czech Republic. The small intestine was cut, rinsed using phosphate-buffered saline (PBS), and mounted into a Franz diffusion cell with PBS as a recipient medium. About 200 µL of SES/thymol/coumarin-6 (86.668, 13.266, and 0.066%, respectively) containing emulsion (2 µL SES + 198 µL PBS) was put into the donor compartment of the Franz diffusion cell. The system was tempered to 37 °C during the experiment. The tissue sample was removed after 1 h and froze immediately to −80 °C using n-heptane. Frozen samples were fixed with Tissue-Tek^®^ and cross-sectioned using Cryocut (Leica Microsystems’, Wetzlar, Germany). Tissue slides of the thickness of 15 µm were transferred to the glass slide and fixed with ice-cold acetone. The nuclei of the samples were stained using DAPI, and the samples were observed using a confocal microscope. 405 nm laser was used to visualize the nuclei, and a 488 nm laser line was used to visualize coumarin-6.

## 3. Results and Discussion

### 3.1. Evaluation of Prepared SES

The first step was to evaluate the size of emulsion droplets. The thymol-containing SES was stained using a fluorescent dye, coumarin-6, and consequently visualized using confocal microscopy. This method allows direct particle size measurement in the water without any effect on the particle size, e.g., by additional dilution needed, drying of the sample, or any other procedure of sample manipulation. Figure 1 shows the visualized particles of the emulsion formed from SES after mixing with water. The particle size was determined between 1 and 7 µm using image analysis immediately after mixing SES with water. No significant difference in the size was observed after 1-h interval. The image analysis method of fluorescently stained particles was selected because of the presence of larger particles. Therefore, the sample was not suitable for measurement using dynamic light scattering. The particle size in a given size range corresponds to the intended purpose of promoting penetration into the intestinal wall, thus avoiding the systemic effect and lymphatic delivery typical of nano- and microemulsions [27].

### 3.2. Technological Characteristics of Uncoated Pellets

The next step in developing self-emulsifying thymol pellets was the preparation of pellet cores (uncoated pellets). The assumed animal model (rabbit) and the GIT specifics for their in vivo testing were considered during formulation [28]. Nine samples with the same composition differing in wetting agent type and amount were formulated by the M1 extrusion/spheronization method (Figure 2). Technological parameters of uncoated pellets are summed up in Table 2.

In the M1 set, a higher yield of the desired fraction (0.5–0.8 mm) was obtained for samples where acetic acid (AA) was used as a wetting agent. It is related to the solubility of chitosan in acidic media due to the protonation of amine groups in its structure [29]. It results in increased chitosan’s binding properties and extruded wet mass plasticity. This observation is consistent with previously published data [30]. The prepared pellet cores were sized in 640–718 μm following a die plate perforation diameter exhibiting almost identical, sufficiently high sphericity (0.95–0.97). Pellets’ flowability was expectedly recognized as excellent/good.

The nature of extrusion/spheronization can be a fundamental barrier to incorporating volatile substances. Moreover, solid-state thymol and some structurally similar substances are also known as pellet porogen [31,32,33,34]. Therefore, thymol content was a critical parameter. Due to its volatility and leakage during the pelletization [35], thymol content was always lower, between 28.7 to 36.9 mg/g across M1 samples, compared to the theoretical content (41.4 mg/g). It can be noticed that the samples M1-W110 and M1-AA(0.25)110 exhibited higher thymol content, probably thank to a more compacted mass hurdling evaporation, when using a higher amount of wetting agent. However, their mechanical properties were poor, as demonstrated by the 5.9 and 4.4 % friability results for samples M1-W110 and M1-AA(0.25)110, respectively.

The M1 samples expressed relatively high density (1.39–1.43 g/cm^3^). This parameter strongly influences the transit times through the GIT segments. High-density pellets exhibit a limited food and fluids intake dependency and significantly prolong the transit time beneficial for reducing the wash-out effect [36,37,38]. Samples with intraparticle porosity ≤ 1.28% showed satisfactorily lower friability ≤ 1.0% compared to the others. According to the scientific literature, friability lower than 1.7% was reported as acceptable. Moreover, low friability is essential for the intended pellet spray coating [39,40]. Regardless of the type, wetting liquid amounts of 90 g were shown as optimal, mainly due to extremely low friability (0.0–0.3 %). Therefore, the M1-W90, M1-AA(0.25)90, and M1-AA(0.5)90 prepared by the M1 extrusion/spheronization method have been shortlisted. The sample M1-W100 also fell within this limit but did not show sufficient process yield. Although samples M1-AA(0.25)90 and M1-AA(0.5)90 expressed almost identical properties to M1-W90, M1-W90 was preferred due to the inertness of water and potential versatility in pellets with various active pharmaceutical ingredients (APIs) [41]. The sample M1-W90 has shown acceptable characteristics also in four-month stability studies. Compared to time 0, there was a slight deterioration in friability in both storage conditions (0.52 ± 0.07 % for 25 °C/60 % RH, 0.90 ± 0.11 % for 40 °C/75 % RH), confirming excellent mechanical durability [39]. During the stability study, the stable thymol content was also proved (−1.33 ± 0.94 % for 25 °C/60 % RH, −1.93 ± 0.27 % for 40 °C/75 % RH).

In the next step, the M1 pellet samples were tested for their in vitro dissolution behavior, as shown in Figure 2a. The results of the similarity factor analysis are shown in Table 3.

It can be noted that all dissolution profiles of the M1 samples showed common features. They exhibited a so-called *burst effect*, where an average of 60.6% of thymol was released during the first 15 min for the W-series, 57.0% for the AA(0.25)-series, and 54.5% for the AA(0.5)-series samples. When exposed to the AA solutions, the *burst effect* can be slightly reduced by chitosan gelling properties, lowering the API diffusion from pellets [30,42]. This observance was the most significant in sample M1-AA(0.5)90. Protonation of -NH_2_ groups in chitosan molecules leads to the formation of swelling cationic gel, which slightly retards thymol release from pellets [43]. In the 120th min, on average, all M1 samples released 87.5% of thymol; in the 360th min, they all released ≥ 100%. Although the differences between W-series and AA-series were found, according to *f*_2_ analysis, all dissolution profiles were considered similar (Figure 2a, Table 3) [44]. The similarity of dissolution profiles has also remained in the M1-W90 stability study. Similarity factor *f*_2_ varied in the interval between 91.62 at 25 °C, 60% RH, and 73.14 at 40 °C, 75% RH after four months compared to the time t = 0.

### 3.3. Technological Characteristics of Uncoated Pellets Prepared by Modified Methods

Based on its excellent properties, sample M1-W90 was selected for further experiments, and its preparation was repeated by simplified methods M2 and M3 (Figure 2). Regarding the mentioned technological characteristics, samples M2-W90 and M3-W90 exhibited a few differences compared to the original M1-W90 (Table 2). The fraction yield of sample M2-W20 was considerably lower than that of M1-W90 and M3-W90, which performed comparably. The sphericity of samples M1-W90, M2-W90, and M3-W90 was extraordinary, but the particle size was slightly higher for simplified methods (705 ± 86 vs. 740 ± 88 and 732 ± 82 µm, respectively). With the shortening of preparation time, the restricted water evaporation changed the total moisture content, thus providing more integrity within the extrusion mass. According to the Ph. Eur. 10, flow properties improved from “good” to “excellent” compared to the M1-W90 sample, most probably triggered by improved particle size, higher sphericity, and pycnometric density [45,46]. Intraparticular porosity was lower in M2-W90 and M3-W90, which also correlates with better flow properties (smooth particles flow freely). The change in the preparation method (M1 vs. M2 or M3) resulted not only in a decent shortening (14 vs. 12 or 5.5 min, regardless of the operator time) of the preparation time but also a significant increase in the thymol content. Compared to the theoretical amount, the thymol content represented 72.5% and 95.7% for the samples M1-W90 and M3-W90, respectively. It can be concluded that reducing the preparation time and incorporating thymol as an SES adsorbed on the NEU structure is an effective aid in limiting the loss of volatile drugs [47,48,49].

The dissolution profiles of the pellet cores M2-W90 and M3-W90 formulated by the more straightforward procedures were similar to M1-W90 (*f*_2_—64.85, 74.62, respectively). The average key points were 61.9%, 89.3%, and 101.6% of liberated thymol in the 15th, 120th, and 360th minutes, respectively (Figure 2b, Table 3).

The M3 method proved advantageous for achieving high API content while maintaining similar dissolution characteristics. Logically, the sample M3-W90 was chosen to formulate final thymol enteric-coated self-emulsifying pellets.

### 3.4. Technological Characteristics of Coated Pellets

To avoid unwanted thymol absorption by the stomach and to support the intestine wall cumulation, M3-W90 pellet cores were spray-coated by Eudragit^®^ L30 D-55, well established in the pharmaceutical industry. It is resistant to the gastric environment and soluble in the small intestine at pH above 5.5 (due to carboxylic group ionization) [50]. The fluid-bed coating could open a space for leakage mainly due to intensive pellet cores movement and temperature setting at 50 °C, corresponding to the temperature of intensive thymol evaporation [51]. Therefore, determination of the thymol content in the final pellets is essential. The efficacy of thymol encapsulation was determined by comparing the thymol content in uncoated (M3-W90) and coated pellet (C-M3-W90) systems (weight gain 50% of origin pellet mass). The coated sample contained 19.6 ± 1.6 mg/g of thymol, corresponding to the 1.01 ± 0.02% loss of thymol during the coating process.

The C-M3-W90 pellets spray-coated by Eudragit^®^ L30 D-55 dispersion were tested for in vitro dissolution behavior in conditions mimicking the GI fluid of the proposed rabbit animal model (Figure 3). The data showed 9.2% of thymol released in acidic conditions during the first 180 min. This satisfactory result can be correlated with the authority standard limiting the drug released in enteric-coated formulations to 10% in 120 min [52].

After the alkalization, the burst liberation of thymol was observed. Disregarding the drug release in an acidic medium, in the first 15 and 30 min, 37.7%, and 49.2% were released, respectively, compared to the 61.5% and 69.7% in uncoated M3-W90 pellets. It was followed by a gradual release of 83.1% of thymol in the 540th min. Noteworthy, continual dissolution was held without the polysorbate 80 addition. Therefore, the release of thymol from the coat-free pellet cores was higher compared to the C-M3-W90 sample. The polysorbate addition was skipped for its long dissolving time. If polysorbate 80 was added in the 180th min, the sample withdrawn in the 195th min would be affected by its incomplete dissolution compared to the following sampling times. However, we hypothesized that the presence of various surfactants in GIT fluids would enhance the complete thymol dissolution [53,54].

### 3.5. Morphology of Uncoated and Coated Pellets

The samples of uncoated M1-W90, M2-W90, M3-W90, and C-M3-W90 pellets were selected for morphological evaluation by SEM. All uncoated self-emulsifying pellets showed a rough surface, which could be a result of the Neusilin^®^ US2 particles (mean size = 106 um, according to technical data sheet [55]), as seen in surface detail (Figure 4A–C). In sample C-M3-W90, the surface smoothening was achieved by the coating with Eudragit^®^ L30 D-55 water dispersion (Figure 4D2). The presence of coating layers is clearly visible in a detailed cross-section photo (Figure 4D3). Due to the length of the coating process, the process had to be divided into three phases, which is reflected in the formation of the separate layers. The shell is approximately 100 µm thick around the entire perimeter of the pellet; thus, the coating is evenly distributed. Under the pellet shell, there is a core with a solidified self-emulsifying system.

### 3.6. Solid-State NMR Spectroscopy

Based on the excellent properties, the samples of the W90-series were subsequently characterized by the strategy developed for mapping the structure of organic phases integrated into mesoporous silica drug-delivery devices [23]. This approach based on a few straightforward solid-state NMR techniques (^1^H MAS, *T*_1_-filtered ^13^C MAS, and ^13^C CP/MAS NMR) is advantageous because it has no limitations regarding concentrations of the active compounds and enables direct discrimination of various organic constitutions. For readers who are not familiar with ss-NMR spectroscopy, a brief description of this strategy is presented below.

At first, consider that ^1^H MAS NMR spectra are strongly receptive to the changes in molecular dynamics. Broad ^1^H MAS NMR signals indicate rigid organic solids in the crystalline or amorphous state, whereas narrowing of the signals indicates released motions. The resolved ^1^H MAS spectra with signals linewidth of ca. 500–50 Hz reflect soft amorphous phase or gels, whereas diluted solutions produce narrowed signals below 50 Hz. Similarly, the *T*_1_-filtered ^13^C MAS NMR spectra measured with a short repetition delay preferentially detect signals of mobile segments. If the signals are narrow, the corresponding molecules undergo fast motions. The signal broadening observed in ^13^C MAS NMR spectra indicates reduced molecular dynamics and the existence of a broad distribution of molecular assemblies typical of amorphous phases. Complementarily, rigid fractions are then preferentially detected in ^13^C CP/MAS NMR spectra. If these signals are narrow, the corresponding molecules are well ordered in crystalline zones. If the signals are broad, the corresponding molecules adopt the disordered nature of amorphous solids.

Specifically, the ^13^C MAS and ^13^C CP/MAS spectra recorded for the samples of the W90-series are shown in Figure 5. In all cases, a direct comparison of ^13^C CP/MAS and ^13^C MAS spectra (above and below, respectively) is presented. When focusing on the soluble organic phase (thymol and SES) resonating mainly in two spectral regions, 180–110 and 50–20 ppm, no significant spectroscopic differences between the pellet cores prepared by different procedures (M1, M2, M3) were apparent by a simple visual inspection of the spectra. Specifically, the ^13^C MAS NMR spectra of all the prepared systems M1, M2, and M3 are dominated by the narrow signals of thymol and SES. In contrast, the corresponding ^13^C CP/MAS NMR spectra contain only weak, barely detectable signals in these spectral regions. This finding thus unambiguously confirms that thymol and SES molecules are, in the majority, highly mobile, forming basically homogenous liquid films/phases, which are incorporated on the surface (internal or external). In addition, the spectral patterns in the range from 120 to 60 ppm detected in ^13^C CP/MAS NMR spectra, which are identical for all the prepared pellet cores, show that the polysaccharides components (MCC, chitosan) are unaffected by different processing. Thus, different preparation protocols (M1, M2, and M3) have no or very limited impact on the structure of the prepared pellet cores.

However, a more detailed inspection of the ^13^C MAS NMR spectra revealed certain differences between the prepared systems. Specifically, thymol signals exhibit slight broadening depending on the preparation procedure used. Whereas the signals of aromatic protons at ca. 140–100 ppm are slightly broader for the system M1 reaching ca. 90 Hz, the corresponding signals detected for the systems M2 and M3 are apparently narrower (ca. 50 Hz). This phenomenon indicates slightly reduced molecular dynamics of thymol in the M1 system. It can be explained by greater adhesion of thymol to the carrier surface in M1 pellets and may indicate a higher degree of non-bonding interactions between the surface and thymol.

This finding is also consistent with the ^13^C CP/MAS NMR spectra in which we can detect the weak signals of an immobilized surface-anchored fraction of dissolved thymol and SES molecules in addition to the strong signals of insoluble polysaccharides. This fraction was unambiguously found for M1 pellets (Figure 6A), in which ^13^C CP/MAS NMR spectrum, the signals of thymol aromatic carbons and SES aliphatic units are clearly apparent at ca. 120 ppm and ca. 30–20 ppm, respectively. When applied to the time-shorter procedure M3, the amount of the immobilized surface-anchored fraction is a bit reduced. In addition, in this case, the surface-immobilized fraction is exclusively formed by SES molecules (Figure 6B). In this regard, it is also worthy to note that the surface-immobilized fraction occurs only in the systems containing polysaccharide components. Without polysaccharides, for instance, in the system of Neusilin^®^ + SES + thymol, the immobilized fractions of SES and thymol molecules were not detected (Figure 6C). In this case, the carrier surface is only partly occupied by the molecules of propylene glycol (PG). This finding thus indicates a positive effect of polysaccharide molecules on the long-term stabilization of self-emulsifying pellets; however, it is impossible to clearly distinguish from the experiment which of the present polysaccharides is responsible for this effect.

The surface-induced slowing down of the molecular dynamics is further demonstrated in ^1^H MAS NMR spectra, in which we see gradual signal broadening (Figure 7). For the simple solution of thymol in SES, the signals of aromatic protons of thymol at ca. 7 ppm are sufficiently narrow to show the typical *J*-coupling multiplet pattern. The signal resolution slightly deteriorates by incorporating the SES-thymol mixture on the Neusilin surface. However, for the M1 pellet system, the signal broadening is so progressive that we can even see coalescence of aromatic signals. Such a coalescence then confirms a substantial increase in the strength of intermolecular interactions, which probably involve thymol hydroxyl groups -OH as indicated by a disappearance of the corresponding signal at ca. 5 ppm.

Trying to get a deeper insight into the extent and strength of surface interactions of thymol and SES molecules, we focused on analyzing ^1^H-^1^H MAS NMR correlation spectra. In this regard, Figure 8 displays typical spectra recorded for the model SES-thymol mixture and the pellet cores M1 and M3. When looking at them, one can immediately see substantial differences in the recorded correlation patterns. Primarily, the absence of strong correlation signals observed for the model solution of thymol in SES (Figure 8A) clearly reflects rapid molecular isotropic motions, which nearly entirely suppresses the coherent ^1^H-^1^H spin-diffusion polarization transfer. However, when incorporated on the carrier surface, the molecular motions of SES and thymol become more hindered and at least partially anisotropic. Under such conditions, the directional ^1^H-^1^H dipolar interactions become active and allow internuclear ^1^H-^1^H spin-diffusion polarization transfer, which is reflected by much stronger off-diagonal correlation signals (Figure 8B and Figure 9C, compare the intensity of the signals on the horizontal red lines). Further, two types of off-diagonal correlation signals can be distinguished in the recorded ^1^H-^1^H MAS NMR spectra.

One type reflects correlations between atoms of the same molecules. Typically, we see these correlations for aromatic protons of thymol at ca. 7 ppm (Figure 8B). These signals then reflect inter- as well as intra-molecular dipolar contacts and seem to dominate the ^1^H-^1^H MAS NMR correlation spectrum of M3 pellet cores. The second type of correlation signals then corresponds to the intermolecular dipolar contacts between different types of molecules only. These correlation signals indicating tight contacts between thymol and SES molecules are clearly apparent in the M1 system (Figure 8C). In this case, the polarization ^1^H-^1^H spin-diffusion transfer between different types of molecules is so efficient that magnetization of certain diagonal signals is completely consumed, as documented by a dramatic decrease of the corresponding signal intensities. The enhanced efficiency of intermolecular transfer in the M1 system is best demonstrated on the slice through the ^1^H-^1^H MAS NMR correlation spectrum (Figure 8C, compare the intensity of the signals on the horizontal red lines).

In summary, the performed ^1^H-^1^H MAS NMR correlation experiments definitely confirmed the presence of surface-immobilized liquid films formed by the solution of thymol in SES located on the surface of pellet cores. Due to the restricted molecular dynamics on the carrier surface, the intermolecular interactions between thymol and SES molecules are highly active, which leads to the formation of specific thymol-SES molecular clusters. The resulting surface film thus exhibits certain structural anisotropy with the preferential orientation of individual molecules. The extent of intermolecular interactions is considerably enhanced or directly initiated by the presence of polysaccharide molecules and is further controlled by the preparation procedure.

### 3.7. Ex Vivo Study

Due to the unique complex effect, thymol is a potential candidate for treating inflammatory or infectious diseases of the small intestine and colon mucosa. In this study, ex vivo experiments confirmed the cumulation of hydrophobic active substances at the site of its action in the intestinal mucosa when administered using the developed SES. For this reason, the SES was stained using coumarin-6 as described above. Figure 9 shows the cumulation of coumarin in the intestinal mucosa co-delivered with thymol using SES. Coumarin was selected to visualize the transport of hydrophobic compounds from the developed SES system into the target tissue. The concentration of hydrophobic compounds in the villi of the intestinal epithelium is observable after 1-h incubation of SES with the intestine. It confirms the transport of hydrophobic active compounds through the mucus layer (visible in the bottom part of Figure 9) and its successful delivery to enterocytes. We assume that the final system’s design will further enhance thymol cumulation in the mucosa. An overall positive impact on the interstitial mucosa can be expected from the mucoadhesive polymer chitosan, particularly the promotion of penetration, antioxidant effect, anti-inflammatory action, or normalization of the intestinal microbiota [56].

## 4. Conclusions

An easily reproducible and time-saving method for preparing thymol self-emulsifying pellets was optimized. Although some technological parameters (fraction yield, mean particle size, sphericity, etc.,) of the samples prepared by three different methods were satisfactory and mostly comparable, the thymol content varied significantly, and ss-NMR spectroscopy revealed noticeable structural differences between the samples. The ^13^C MAS NMR spectra revealed that the thymol molecules in the M1-W90 sample exhibited slightly reduced molecular dynamics compared to M2-W90 and M3-W90, indicating a higher degree of non-bonding interactions between the carrier surface and thymol. In addition, a surface-anchored fraction of immobilized thymol and SES molecules was detected in the M1-W90 sample by ^13^C/CP MAS NMR experiment, which is consistent with the measurement above. In the case of the pellets with the shortest preparation time (M3-W90), the immobilized surface-anchored fraction was reduced and consisted exclusively of SES molecules. It was found that immobilized SES or thymol fractions were only present in systems containing polysaccharides, indicating a positive effect on the long-term stability of self-emulsifying pellets. We suggest that the impact of a specific polysaccharide on SES or thymol immobilization should be studied in the future, which has not been elucidated in this work. ^1^H MAS NMR spectra confirmed the carrier surface-induced slowing down of the molecular dynamics by comparing the spectra of the SES-thymol solution, its intermediate with Neusilin^®^, and the pellets M1-W90 and M3-W90. An advanced spectroscopic experiment, 2D ^1^H-^1^H NOESY MAS NMR, was used to compare M1-W90 and M3-W90 samples. The weak correlation signals observed for the model thymol solution in SES and their appearance after solidification by M1 and M3 methods indicate a change from fast isotropic motion to a more hindered and at least partially anisotropic motion after solidification. This underlines the presence of surface-immobilized liquid films. However, sample M3-W90 was selected for film coating as it exhibited the highest thymol content along with excellent technological properties. Coated C-M3-W90 was successfully tested in continual dissolution, mimicking stomach passage to the intestine. In the ex vivo study, coumarin-6 was co-administered with thymol in the form of SES. This final experiment confirmed the accumulation of thymol in the intestinal wall and its penetration into enterocytes. We believe that the results obtained justify conducting an in vivo study in the future.

## Data Availability

A full list of references is compiled and attached to this manuscript.

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
