# Peer review of "Rational Design of Self-Emulsifying Pellet Formulation of Thymol: Technology Development Guided by Molecular-Level Structure Characterization and Ex Vivo Testing"

_pharmaceutics, 2022, doi:10.3390/pharmaceutics14081545_

Round 1
Reviewer 1 Report
On behalf of Pharmaceutics, I reviewed the article entitled Rational Design of Self-Emulsifying Pellet Formulation of Thymol: Technology Development guided by Molecular-Level Structure Characterization and Ex-vivo Testing by Macku et al. The methods employed are properly described and the comments provided are well supported by the results. The section of the Results and discussion is more properly written, while the abstract and the Introduction need some adjustments.
Specific comments
· The Abstract is too general in some parts: e.g. for what thymol is used for in therapy is not reported, data on size, permeation, dissolution are not provided in the results
· Line 43, the statement should be supported by reference and should be clarified. What did the Authors mean for “recent complex information”?
· Actually, the Introduction is not well structured. For instance, from line 52 to line 60 the Authors describe the GI diseases and before and after this paragraph they speak about thymol. They should not jump from one topic to another and then return to the first one.
· Line 67, please explicit S-SEDDS
· Line 87, the self-citations are redundant, please report only a few from your lab and add another from other colleagues or a review.
· Line 89, why to mention buccal films?
· Authors introduce the ss-NMR technology at line 84 and again at line 102, in the middle they describe methods that are not used in the present work, so please delete this part or reduce and transfer it.
· Are there in the literature other drug delivery systems loaded with thymol? Can you describe and discuss them just a little as to eventually highlight the pros and cons with respect of your approach?
· Line 114, “We were able to correlate their targeted formulation with optimized properties”, this sentence is not clear please reformulate it
· Line 277, From whom were the animals purchased?
Author Response
Response to Reviewer 1 Comments
We thank the reviewer for his/her valuable comments. All reviewer’s comments and suggestions were taken into consideration and our contribution has been revised accordingly. We hope that the performed changes have improved clarity of the presented results. Thank you very much.
Comment 1.: “…The Abstract is too general in some parts: e.g. for what thymol is used for in therapy is not reported, data on size, permeation, dissolution are not provided in the results…”
Authors reply: On the basis of the reviewer’s requirement we tried to rewrite the Abstract section. Specifically, we removed too general phrases and added some required details. We hope that the current version of the Abstract is more informative and more explicit for readers (see page 1, lines 15-30).
Comment 2.: “…Line 43, the statement should be supported by reference and should be clarified. What did the Authors mean for “recent complex information”?...”
Authors reply: The reviewer is right; the reference was missing, and the term “complex information” was a bit confusing. Therefore, we modified this sentence accordingly and added a new citation [ref. [2]; page 2; lines 40-41].
[2] Kumar, A.; Choudhir, G.; Shukla, S.A.; Sharma, M.; Tyagi, P.; Bhushan, A.; Rathore, M. Identification of phytochemical in-hibitors against main protease of COVID-19 using molecular modeling approaches. J Biomol Struct Dyn. 2020, 39(10),3760-3770. doi: 10.1080/07391102.2020.1772112
Comment 3.: “…Actually, the Introduction is not well structured. For instance, from line 52 to line 60 the Authors describe the GI diseases and before and after this paragraph they speak about thymol. They should not jump from one topic to another and then return to the first one…”
Authors reply: Yes, the reviewer is right. On the basis of the reviewer’s comment we reorganized the introductory section accordingly [page 2; lines 42-64].
Comment 4.: “…Line 67, please explicit S-SEDDS…”
Authors reply: Regarding the reviewer’s comment, redundant shortcut was deleted and the phrase “solid self-emulsifying system” was replaced (see Introduction, page 2, line 65).
Comment 5.: “…Line 87, the self-citations are redundant, please report only a few from your lab and add another from other colleagues or a review…”
Authors reply: OK, the reviewer is right again. On the basis of the reviewer’s objection we removed our citations [refs. 20, 21, 22 and 24], (in the original version) and added a new citation referring to the review article by Marco Geppi ([ref. 26], see page 2; line 92)
Comment 6.: “…Line 89, why to mention buccal films?...”
Authors reply: We agree with the reviewer that the original version was not precise. Therefore, we modified/rewrote this section and removed the reference to buccal films (see page 2 lines 93-98)
Comment 7.: “…Authors introduce the ss-NMR technology at line 84 and again at line 102, in the middle they describe methods that are not used in the present work, so please delete this part or reduce and transfer it…”
Authors reply: Also, in this case we agree with the reviewer. Therefore, we considerably rewrote the corresponding paragraphs (see page 2 lines 93-98).
Comment 8.: “…Are there in the literature other drug delivery systems loaded with thymol? Can you describe and discuss them just a little as to eventually highlight the pros and cons with respect of your approach?...”
Authors reply: Thank you very much for this comment. We added (see page 2 line 72-78) into the Introduction and one more reference [20].
[20] Rassu, G.; Nieddu, M.; Bosi, P.; Trevisi, P.; Colombo, M.; Priori, D.; Manconi, P.; Giunchedi, P.; Gavini, E.; Boatto, G. Encapsulation and Modified-Release of Thymol from Oral Microparticles as Adjuvant or Substitute to Current Medications. Phytomedicine 2014, 21, 1627–1632. https://doi.org/10.1016/j.phymed.2014.07.017
Comment 9.: “…Line 114, “We were able to correlate their targeted formulation with optimized properties”, this sentence is not clear please reformulate it…”
Authors reply: We agree with the reviewer that the original sentence was a little bit clumsy. We have therefore modified this sentence (see page 3 lines 105-106).
Comment 10.: “…Line 277, From whom were the animals purchased?...”
Authors reply: The rabbits were purchased in AnLab, s.r.o., Prague, Czech Republic. (https://www.anlab.cz/). This information has been added (see page 7, line 267).
Reviewer 2 Report
The growing demand for processing natural lipophilic and often volatile sub-stances led researchers to include solid-state nuclear magnetic resonance inspection (ss-NMR), guiding the rationalized development of gastro-resistant pellets with a thymol self-emulsifying system for treating intestinal diseases. In this work, author designed an oral system (Uncoated thymol with a self-emulsifying system) to prevent undesirable thymol stomach absorption and promote its intestinal cumulation. The optimal excipients composition processed by different methods provided comparable pellets with varying thymol content, increasing with shorter process time, and similar in-vitro dissolution behavior. Although only minor differences were found at the first inspection, a detailed ss-NMR study revealed remarkable structural differences across samples differing in the preparation procedure concerning system stability and internal interaction. The ex-vivo experiment confirmed thymol cumulation in enterocytes. Overall, the suggested formulation and methodology may be promising for using other lipophilic volatiles with potential benefits in treating intestinal diseases. I think this work is worthy to publish in Pharmaceutics after fully addressing the following concerns.
1. The fundamental problem is that monoterpene molecules are predominantly hydrophobic, with poor oral bioavailability. It is common sense that the most of monoterpene molecules are hydrophobic, however, more literature should be cited here to highlight this challenge. In addition, “…According to available literature, thymol is only partially absorbed mainly from the stomach, resulting in a 16% bioavailability…”, where is the available references?
2. From SEM images, it shows that the outer layer of the specimen is very dense, will this structure affect the practical performances (e.g., drug release…) of coated pellets?
3. Chitosan as a kind of polysaccharide enables protonation and becomes cationic macromolecules. So, in the practical application, the drug molecules will have specific interaction with chitosan? Some discussions should be added in the Results.

Author Response
Responses to the Comments of Reviewer No. 2
We thank the reviewer for the valuable comments. All reviewer’s comments and suggestions were taken into consideration and our contribution has been revised accordingly. We hope that the performed changes have improved clarity of the presented results. Thank you very much.
Comment 1.: “The fundamental problem is that monoterpene molecules are predominantly hydrophobic, with poor oral bioavailability. It is common sense that the most of monoterpene molecules are hydrophobic, however, more literature should be cited here to highlight this challenge. In addition, “…According to available literature, thymol is only partially absorbed mainly from the stomach, resulting in a 16% bioavailability…”, where is the available references?”
Authors reply: The reviewer is right; the corresponding references were missing in the original version of our manuscript. Our mistake. Therefore, we tried to find and add new references as required. However, data on the bioavailability of thymol or other monoterpenes are almost unavailable. The studies we found rather deal with absorption, deposition and distribution. However, nowhere bioavailability is quantified explicitly, except for the article cited below which gives 16% for thymol (Kohlert et. Al, 2002) [ref 14]. This reference is now explicitly mentioned in the revised version of the manuscript (see Introduction, page 2, lines 59-60).
In addition, according to the reviewer’s suggestion we added references [ref 12], [ref 13] regarding monoterpenes properties to underline the need of appropriate formulation. This part of the manuscript was also modified (see Introduction, page 2, lines 55-64).
References:
[12] Lorenzo, J. M.; Mousavi Khaneghah, A.; Gavahian, M.; MarszaÅ‚ek, K.; EÅŸ, I.; Munekata, P. E. S.; Ferreira, I. C. F. R.; Barba, F. J. Understanding the Potential Benefits of Thyme and Its Derived Products for Food Industry and Consumer Health: From Extraction of Value-Added Compounds to the Evaluation of Bioaccessibility, Bioavailability, Anti-Inflammatory, and Antimicrobial Activities. Crit. Rev. Food Sci. Nutr. 2019, 59, 2879–2895. https://doi.org/10.1080/10408398.2018.1477730
[13] Salehi, B.; Mishra, A. P.; Shukla, I.; Sharifi-Rad, M.; Contreras, M. del M.; Segura-Carretero, A.; Fathi, H.; Nasrabadi, N. N.; Kobarfard, F.; Sharifi-Rad, J. Thymol, Thyme, and Other Plant Sources: Health and Potential Uses: Thymol, Health and Potential Uses. Phytother. Res. 2018, 32, 1688–1706. https://doi.org/10.1002/ptr.6109
[14] Kohlert, C.; Schindler, G.; März, R. W.; Abel, G.; Brinkhaus, B.; Derendorf, H.; Gräfe, E.-U.; Veit, M. Systemic Availability and Pharmacokinetics of Thymol in Humans. J. Clin. Pharmacol. 2002, 42, 731–737. https://doi.org/10.1177/009127002401102678
Comment 2.: “From SEM images, it shows that the outer layer of the specimen is very dense, will this structure affect the practical performances (e.g., drug release…) of coated pellets?”
Authors reply: The spray-coated layer is very dense due to the coating process's length and relatively high coating mass (w/w(cores) 50 %). The large surface area of pellets requires x-fold higher amount of coating material compared to the tablets, to achieve desired lag-time Unfortunately, the lower amount of water dispersion of Eudragit did not lead to the desired lag time. The thick and dense layer delayed the drug release as shown in Figure 3. Only 10 % of thymol was released during 180 min period. However, simplifying and shortening the coating process will be a challenge for further investigation. Regarding other parameters, we suggest that coated pellets are likely to be more spherical and less fragile.
Comment 3.: Chitosan as a kind of polysaccharide enables protonation and becomes cationic macromolecules. So, in the practical application, the drug molecules will have specific interaction with chitosan? Some discussions should be added in the Results.
Authors reply: Regarding the reviewer's question, we consider a possible indirect interaction between thymol and chitosan. This interaction, which we describe as indirect, occurs as a result of the small thymol molecules being spatially hindered by the large polysaccharide molecules. However, by given experiments, we did not distinguish whether microcrystalline cellulose or chitosan was responsible for the observed behavior. Nevertheless, a weak Van der Waals and hydrogen-bond interactions involving thymol -OH groups and chitosan NH2 units should be considered as highly probable.
Additionally, when exposed to acidic conditions, the -NH2 groups of chitosan are protonated, which may lead to the formation of “gel-like-barrier” slowing thymol release. In presence of anionic molecule, for example TPP (sodium triphosphate), the cross-linking of polysaccharide would occur, which would probably lead to even more dense barrier slowing thymol release. In these cases, we would describe chitosan specific interaction.
As required by the reviewer the discussion in the Results was extended (see Results, page 10, lines 360-362). In this consequence the reference [ref. 47] was added.
References:
[47] Nilsen-Nygaard, J.; Strand, S.; Vårum, K.; Draget, K.; Nordgård, C. Chitosan: Gels and Interfacial Properties. Polymers 2015, 7, 552–579. https://doi.org/10.3390/polym7030552